# Realizing self-sustained biomass gasification in a lab-scale downdraft reactor for compact CHP applications

**Takanori Itoh**[1], **Shota Hanabusa**[2], **Kazunori Iwabuchi**[1*]

1 Research Faculty of Agriculture, Hokkaido University, Sapporo, Hokkaido, Japan, 2 Graduate School of Agriculture, Hokkaido University, Sapporo, Hokkaido, Japan

\* iwabuchi@agr.hokudai.ac.jp

## Abstract

Miniaturizing biomass gasifiers is key to enabling household-scale combined heat and power (CHP) systems integrated with solid oxide fuel cells (SOFCs), yet achieving self-sustained gasification at this scale is challenging due to severe heat losses. This study evaluates the self-sustaining potential of a lab-scale moving-bed downdraft gasifier using charcoal, across an equivalence ratio (ER) range of 0.47–0.65. Stable operation was achieved at ER = 0.47–0.60, with oxidation zone temperatures reaching 800–1000 °C. As ER increased, product gas performance improved, with CO and $H_2$ concentrations of 15% and 5% at ER = 0.60, yielding a lower heating value of 2.60 MJ/Nm³, carbon conversion efficiency of 74.4%, and cold gas efficiency of 48.4%. Performance remained below literature values, primarily due to limited reduction zone temperatures (~600 °C), likely caused by incomplete oxygen conversion (residual $O_2$: 5.9%) and significant heat losses (~30.5%). Thermal analysis showed that reducing residual $O_2$ below 1% could theoretically raise temperatures by nearly 1000 °C. These findings demonstrate the feasibility of lab-scale autothermal gasification and identify oxygen utilization and thermal management as key levers for future ultra-small-scale CHP applications.

## 1. Introduction

Biomass is increasingly recognized as a carbon-neutral and dispatchable renewable energy source, offering significant potential for decentralized energy systems. However, its relatively low energy density and geographically dispersed nature lead to high costs for feedstock collection, transportation, and preprocessing, often impeding economic viability in large-scale power applications [1]. To overcome these challenges, small-scale combined heat and power (CHP) systems based on biomass gasification have been proposed, offering the potential to reduce fuel transportation costs and enable localized energy use with high system efficiency. Particularly in Japan, household-level CHP systems powered by natural gas or liquefied petroleum

**Data availability statement:** All relevant data are within the manuscript and its Supporting information files.

**Funding:** This study was supported by funding from Tanigurogumi Corporation and the Japan Society for the Promotion of Science (JSPS KAKENHI, JP23K23727).

**Competing interests:** This research was partially funded by Tanigurogumi Corporation, Japan. The authors affirm that the study design, data collection, analysis, and interpretation of results were conducted independently, without external influence. All other authors declare no known competing financial interests or personal relationships that could have influenced the work reported in this paper.

gas and solid oxide fuel cells (SOFCs) have already been implemented, with typical electrical outputs of 200–700 W [2]. If biomass can be utilized as a feedstock in such systems, it would represent one of the most efficient pathways for practical biomass energy utilization.

Biomass gasification is a thermochemical process that partially oxidizes biomass at 800–1000 °C to produce a combustible gas mixture rich in CO and $H_2$. Among various reactor types, the downdraft gasifier is especially suitable for small-scale CHP applications due to its low tar content and high fuel conversion efficiency [3,4]. Small-scale gasification systems ranging from 50 to 200 $kW_{th}$ have been demonstrated in various studies. For example, a 75 $kW_{th}$ downdraft gasifier was reported to produce gas with a lower heating value (LHV) of 4.15 MJ/$Nm^3$ [5], while a 50 $kW_{th}$ system with staged air supply achieved an LHV of 4.54 MJ/$Nm^3$ and a cold gas efficiency (CGE) of 68% [6]. Two-staged systems, combining pyrolysis and gasification units, have demonstrated even higher performance with LHVs exceeding 6 MJ/$Nm^3$ and CGEs of up to 93% [7,8]. Nevertheless, these systems still operate in the tens-of-kilowatts range, which is unsuitable for direct household application. Lab-scale gasifiers, on the other hand, have been studied primarily to understand fundamental gasification behavior under controlled conditions, often using external heaters to maintain high temperatures. For example, [9] investigated an updraft reactor with a 34 mm diameter, while [10] used a fluidized bed reactor with a 75 mm diameter. These studies, though insightful, relied heavily on external heating and did not address the feasibility of self-sustained operation under heat-loss-dominated conditions.

To advance biomass-based household CHP, further downscaling to the sub-kilowatt level (200–700 W) is essential. Assuming a charcoal LHV of 25 MJ/kg, gasification efficiency of 80% [11], and SOFC electrical efficiency of 40% [12], the required feedstock input is estimated to be 0.09–0.32 kg/h. Based on a typical hearth loading rate of 100–300 kg $m^{-2}$ $h^{-1}$, this corresponds to a gasifier diameter of approximately 20–63 mm. Accordingly, this study employs a 46 mm inner diameter (50 mm outer diameter) downdraft gasifier, which falls within the laboratory scale.

A key challenge at this scale is significant heat loss, which can compromise the reactor temperature and limit gasification performance. While equivalence ratio (ER) is a well-known operational parameter influencing both gas quality and temperature, conventional downdraft gasifiers typically operate optimally at ER values between 0.2 and 0.4 [13,14]. However, in lab-scale systems with greater thermal losses, higher ER may be required to achieve sufficient combustion and maintain a self-sustaining temperature regime. Therefore, this study aims to investigate whether self-sustained operation is achievable in a compact, non-heated laboratory-scale downdraft gasifier by optimizing the ER. The effects of ER on temperature profiles, syngas composition, and conversion efficiencies are evaluated to assess the feasibility of such systems as candidates for ultra-small-scale biomass-based CHP. The novelty of this study lies in demonstrating practical self-sustained operation in a reactor of only 46 mm inner diameter, without external heating, and under conditions relevant for real-world applications. The findings provide valuable insight for the future development of compact, efficient, and autonomous biomass gasification systems for household energy needs.

## 2. Materials and methods

### 2.1. Materials

Commercial charcoal derived from a mixture of hardwoods and softwoods was used as the feedstock and its fuel properties are shown in Table 1. To prevent clogging within the gasification reactor, the biochar particles were sieved to obtain a size fraction between 2.00 mm and 4.75 mm. The feedstock was oven-dried at 105 °C for 24 hours prior to use.

### 2.2. Gasification system

An overview of the gasification system is shown in Fig 1. The gasifier consisted of a quartz glass tube with an inner diameter of 46 mm, an outer diameter of 50 mm, and a height of 300 mm. To ensure thermal insulation, the outer wall of the

**Table 1. Fuel properties of the feedstock.**

| Items | Values |
|---|---|
| Volatile matter (%db) | 35.41 |
| Fixed carbon (%db) | 60.71 |
| Ash (%db) | 3.88 |
| C (%daf) | 72.17 |
| H (%daf) | 3.43 |
| N (%daf) | 0.60 |
| O (%daf) | 23.80 |
| Higher heating value (MJ/kg) | 26.48 |
| Lower heating value (MJ/kg) | 25.75 |

db: dry basis; daf: dry ash-free.

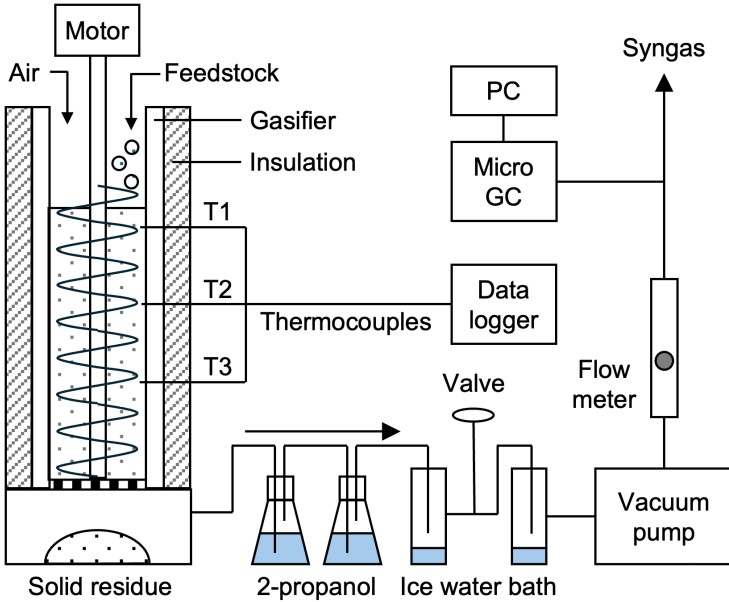

**Fig 1. Schematic diagram of lab-scale gasification system.**

reactor was wrapped with a 15 mm-thick alumina blanket, with a thermal conductivity of 0.18 W m$^{-1}$ K$^{-1}$ at 800°C and 0.36 W m$^{-1}$ K$^{-1}$ at 1200 °C.

The feedstock was manually fed from the top of the reactor at a rate of 0.2 kg/h. Feeding and discharge were performed using a stainless steel auger with a blade diameter of 35 mm installed inside the reactor. Reactor temperatures were measured at three positions along the inner wall: 60 mm (T3), 105 mm (T2), and 150 mm (T1) above the grate, using K-type thermocouples.

Ambient air at 25 °C was used as the gasifying agent and was drawn into the reactor via a vacuum pump. The airflow rate was controlled using a valve and measured with a flow meter. After exiting the lower part of the reactor, the product gas was passed through a tar trap (2-propanol) and a cold trap (ice water bath) to remove particulates, tar, and water vapor. The concentrations of $O_2$, CO, $CO_2$, $H_2$, $CH_4$, and $N_2$ in the product gas were analyzed using a micro gas chromatograph (990 Micro GC, Agilent Technologies).

## 2.3. Gasification operation

Prior to operation, 60 g of biomass feedstock was loaded into the reactor (to a bed height of 150–160 mm from the grate), and ignition was initiated by introducing a heat source from the top. After confirming ignition and stable combustion within the reactor, continuous feeding of feedstock was started. The airflow rate was adjusted using the valve and flow meter to achieve the desired ER, which was set in the range of 0.47–0.65. This ER range was determined based on preliminary experiments, where it was found to allow self-sustained operation of the gasifier without external heating. The auger was operated intermittently to maintain the feedstock bed height between 150 and 160 mm.

Char residues that passed through the grate were collected every 15 minutes. The steady-state condition was defined as the point at which the reactor temperature stabilized and all the initially loaded feedstock had been discharged. Gas sampling was conducted at 5-minute intervals after reaching steady state. The gasifier was operated for an additional 1 hour under steady-state conditions, and the average values of gas composition measured during this period were used for analysis.

## 2.4. Analyses

Proximate, ultimate, and calorific analyses were conducted to characterize the fuel properties. Ash content was determined by a two-step thermal treatment: oven-dried samples were first heated at a rate of 4.5 °C/min to 250 °C and held for 60 minutes, followed by heating at 10 °C/min to 550 °C and maintained for 120 minutes. The remaining mass after this treatment was taken as the ash content. Volatile matter (VM) content was measured based on the mass loss after heating the oven-dried subsamples at 950 ± 20 °C for 7 minutes using an ICKV electric furnace (Ishizuka Electronic, Tokyo, Japan), in accordance with ASTM E872-82 [15]. Fixed carbon (FC) content was calculated by difference using the equation: FC = 100 − VM − Ash. Elemental analysis of carbon (C), hydrogen (H), and nitrogen (N) was conducted using a CE-440 elemental analyzer (Exeter Analytical, North Chelmsford, MA, USA). The oxygen (O) content was calculated by difference: O = 100 − C − H − N. The higher heating value of the feedstock (HHV$_{feedstock}$) was measured using a bomb calorimeter (OSK 200, Ogawa Sampling, Saitama, Japan). The lower heating value (LHV$_{feedstock}$) of the feedstock was calculated from the HHV$_{feedstock}$ using the following equation [11]:

$$LHV_{feedstock} = HHV_{feedstock} - L_v \left( \frac{9H}{100} \right)$$

(1)

where $L_v$ is the latent heat of vaporization of water at 25 °C (2441 kJ/kg), and $H$ is the hydrogen content of the feedstock (3.30%).

The LHV of the product gas, CGE, and CCE were calculated based on the product gas composition, following the methodology described previously [16]. All experiments were performed in duplicate to ensure reproducibility, and the reported values represent the average of the repeated runs.

## 2.5. Thermal analysis and energy-balance calculations

To interpret the temperature behavior of the lab-scale downdraft gasifier, three simplified calculations were conducted: (i) a theoretical temperature rise associated with improved oxygen utilization, (ii) estimated heat-loss fraction from a simplified energy balance, and (iii) a first-order estimate of the temperature gain expected from scaling the reactor diameter. Unless otherwise noted, all calculations were performed on a molar basis of dry product gas and used measured gas compositions and temperatures as inputs.

### 2.5.1. Theoretical temperature rise by enhanced oxygen utilization.

A theoretical temperature rise was estimated to evaluate how much higher the reactor temperature could have been if oxygen utilization were improved. Specifically, the calculation assumes that residual $O_2$ in the product gas is additionally consumed by reaction with char, resulting in extra heat release. The reduction of residual oxygen from ($x_{O_2,obs}$) to a target molar fraction ($x_{O_2,target}$) corresponds to an additional oxygen amount (per 1 mol of dry product gas):

$$\Delta n_{O_2} = x_{O_2,obs} - x_{O_2,target} \tag{2}$$

As an upper-bound estimate, all additional $O_2$ was assumed to react with char to form $CO_2$:

$$C(s) + O_2 \rightarrow CO_2 \tag{3}$$

The additional heat release ($Q_{additional}$ [kJ/mol]) is then calculated from the enthalpy of formation of $CO_2$ ($\Delta H_{f,CO_2} = -394$ kJ/mol):

$$Q_{additional} = \Delta n_{O_2} \left| \Delta H_{f,CO_2} \right| \tag{4}$$

The corresponding adiabatic temperature increase ($\Delta T_{O_2}$ [K]) was estimated by:

$$\Delta T_{O_2} = \frac{Q_{additional}}{C_{p,gas}} \tag{5}$$

where the heat capacity of product gas ($C_{p,gas}$ [kJ mol$^{-1}$ K$^{-1}$]) was calculated by a molar-fraction-weighted sum:

$$C_{p,gas} = \sum x_i C_{p,i} \tag{6}$$

The component heat capacities $C_{p,i}$ were taken from standard correlations/databases, and representative values used in this study are provided in S1 Table.

### 2.5.2. Estimated heat-loss fraction from a simplified overall energy balance.

To estimate heat loss from the reactor, a simplified overall energy balance was applied by comparing the enthalpy released by exothermic oxidation reactions with the sensible heat retained by the dry product gas. The reactor was treated as adiabatic except for heat losses to the environment. For heat generation, only CO and $CO_2$ formation were considered. The sensible heat retained by the gas was calculated from the gas composition and temperature using mixture heat capacities ($C_{p,gas}$).

The reaction heat release per mol of dry product gas was estimated as:

$$Q_{reaction} = x_{CO} \left| \Delta H_{f,CO} \right| + x_{CO_2} \left| \Delta H_{f,CO_2} \right| \tag{7}$$

The sensible heat retained by the product gas was calculated as:

$$Q_{sensible} = C_{p,gas} \left( T_{gas} - T_{in} \right) \tag{8}$$

where $T_{gas}$ is a representative gas temperature and $T_{in}$ is the ambient (inlet) temperature (taken as 298 K). The heat-loss fraction was then determined as:

$$\eta_{loss} = 1 - \frac{Q_{sensible}}{Q_{reaction}} \tag{9}$$

**2.5.3. First-order estimate of temperature increase by scaling reactor diameter.** To assess whether increasing the reactor diameter could mitigate heat losses and thereby raise the attainable gasification temperature in a lab-scale reactor, a first-order scaling analysis was performed. The estimated heat-loss fraction $\eta_{loss}$ was converted to a heat-loss power using the molar flow rate of dry product gas $\dot{n}$. The reaction heat-release power was calculated as:

$$\dot{Q}_{reaction} = \dot{n} Q_{reaction} \tag{10}$$

and the heat-loss power was then estimated as:

$$\dot{Q}_{loss} = \eta_{loss} \dot{Q}_{reaction} \tag{11}$$

Assuming similar operating conditions and insulation configuration, the overall heat-loss power was approximated to scale inversely with reactor diameter ($\dot{Q}_{loss} \propto 1/D$). The heat saved by increasing the diameter from $D_1$ to $D_2$ was then estimated and converted to a corresponding temperature increase of the product gas:

$$\dot{Q}_{saved} = \dot{Q}_{loss}(D_1) - \dot{Q}_{loss}(D_2) \tag{12}$$

Finally, the corresponding increase in product-gas tempearture was calculated by:

$$\Delta T_D = \frac{\dot{Q}_{saved}}{\dot{n} C_{p,gas}} \tag{13}$$

## 3. Results and discussion

### 3.1. Gasification temperature

The temporal variation in reactor temperature is shown in Fig 2. Stable operation was achieved when the ER was maintained at 0.47–0.60 (Fig 2a–c). During trials that successfully maintained steady-state operation, the highest temperature was observed at T1 (upper region of the reactor), followed by T2 (middle) and T3 (lower). Overall, as ER increased, temperatures in the reactor also increased. This trend is attributed to the enhancement of combustion in the oxidation zone due to increased oxygen supply, resulting in greater heat generation. In contrast, at ER = 0.65, corresponding to failure to maintain steady-state operation, the temperatures at T2 exceeded that at T1 as the operation progressed (Fig 2d). This reversal indicates a reduction in the feedstock bed height due to excessive consumption, as combustion occurred faster than the feed rate. The results demonstrate that self-sustaining operation of lab-scale gasifier is achievable when suitable ER (0.47–0.60) is set.

The gasification process involves both exothermic oxidation reactions and endothermic reduction reactions. Based on the measured temperature distribution and gas composition (see below), oxidation was assumed to occur around the upper zone (T1), while reduction reactions dominated in the lower zones (T2 and T3). The heat generated by oxidation supported subsequent thermal decomposition and reduction reactions within the reactor. On the other hand, as concerned, the temperatures in the reduction zone were lower than those in previous studies, due to heat loss from the gasifier wall. Previous studies using downdraft gasifiers have reported reduction zone temperatures typically ranging from 800

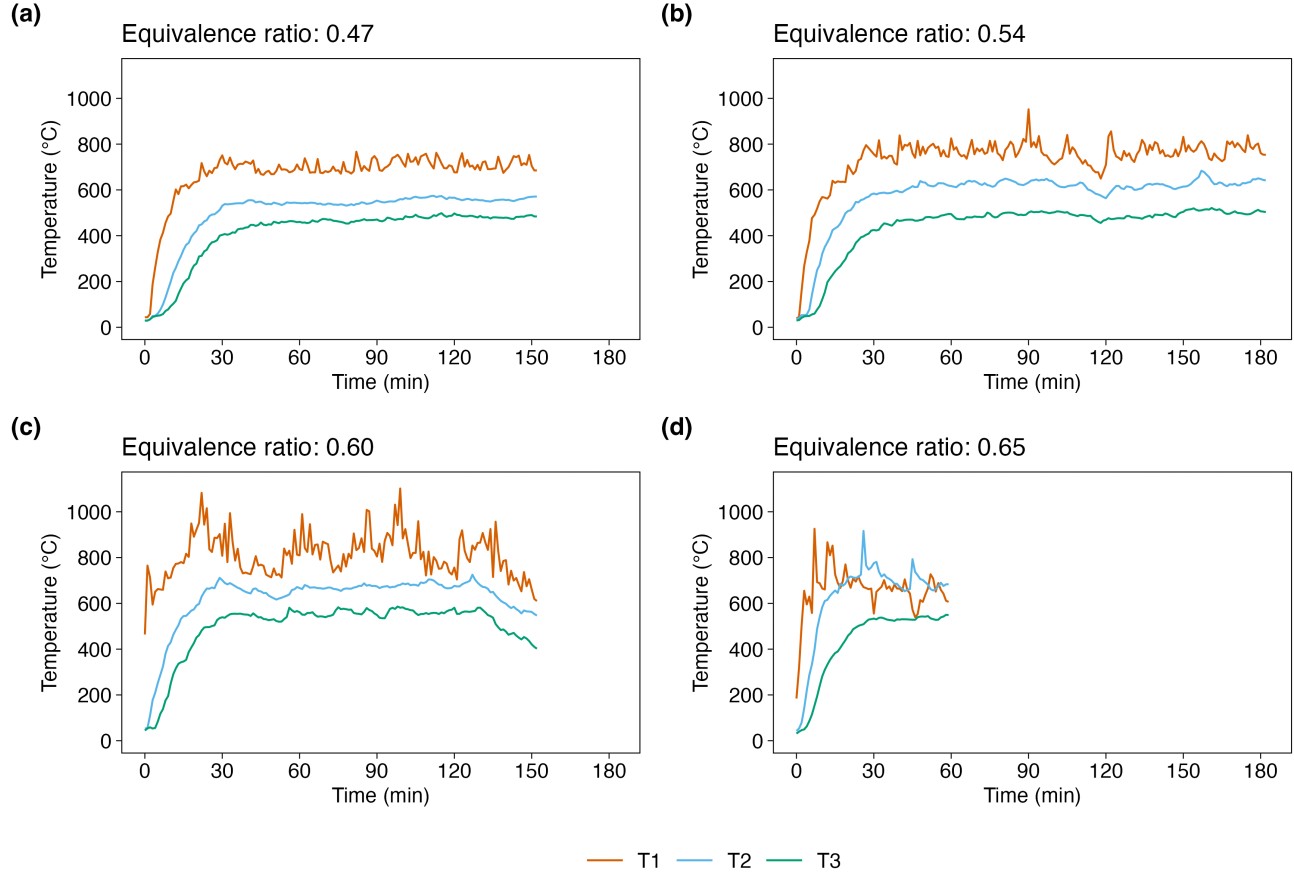

**Fig 2. Temporal profiles of internal reactor temperatures at three vertical positions (T1: upper, T2: middle, T3: lower) under different equivalence ratios (ER = 0.47, 0.54, 0.60, and 0.65).** Stable temperature stratification was observed in (a)–(c), where T1 consistently showed the highest temperature, followed by T2 and T3. In contrast, in (d) at ER = 0.65, T2 and T3 temperatures exceeded T1 over time, indicating a reduction in the feedstock bed height due to excessive consumption.

°C to 900 °C [16–18]. Even at the highest overall gasification temperature of ER = 0.60, the temperature in the reduction zone was about 620 °C (T2), suggesting that this low temperature contributed to the lower gas quality.

### 3.2. Gas composition

The influence of ER on syngas composition is illustrated in Fig 3. As ER increased, CO concentration rose to a maximum of approximately 15 vol%, while $O_2$ and $CO_2$ concentrations decreased to 5.9 vol% and 6.5 vol%, respectively. In contrast, the concentrations of $H_2$ (4.6–5.0 vol%) and $CH_4$ (0.2–0.4 vol%) showed minimal variation across ER values. Notably, under non-steady-state conditions at ER = 0.65, further $O_2$ consumption (4.3 vol%) was observed, and both CO and $H_2$ concentrations increased to 22 vol% and 6.3 vol%, respectively.

Compared to previous studies, which reported CO and $H_2$ concentrations in the range of 20–25 vol% and 13–14 vol%, respectively [16,17,19], the values obtained in this study were significantly lower. This discrepancy is primarily attributed to the lower gasification temperature inside the reactor. The Boudouard reaction ($C + CO_2 \leftrightarrow 2CO$), a key CO-forming reaction, proceeds more readily at high temperatures [20]. For instance, it has been reported that CO concentration increases from ~15 vol% at 600 °C to ~30 vol% at 1100–1300 °C, where CO production becomes thermodynamically dominant [21]. In this study, although increasing ER promoted combustion and char oxidation—as evidenced by the rise in CO and decline

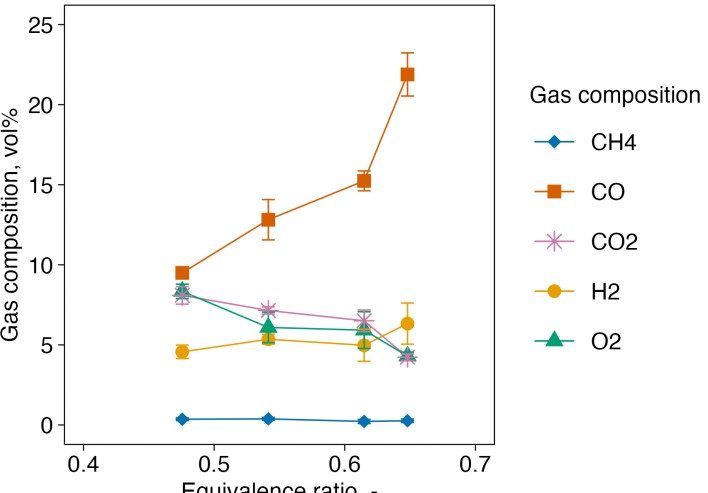

**Fig 3. Effect of equivalence ratio on gas composition.** Data points at ER 0.67 are shown as reference values since it did not achieve steady state.

in $CO_2$—peak temperatures remained within the 800–1000 °C range even at ER = 0.60 (T1), thereby limiting further CO production. In addition to temperature limitation, lab-scale fixed-bed systems can exhibit reduced effective residence time of volatiles and product gas within the high-temperature reaction zone due to limited reactor volume and preferential flow (channeling), which constrains secondary gas-phase reactions (e.g., tar cracking/reforming) and gas–solid reactions [22,23]. In particular, limited secondary reforming/tar cracking can suppress $H_2$ formation, while insufficient gas–solid reaction progress (e.g., char–$CO_2$/steam reactions) can limit additional CO generation. Such hydrodynamic limitations could also contribute to the comparatively low CO and $H_2$ concentrations observed here, even when the maximum bed temperature approaches ~1000 °C.

Interestingly, while high ER is generally associated with CO oxidation at elevated temperatures—leading to a decrease in CO concentration at ER > 0.4 [16,17,19]—such a decline was not observed even at ER = 0.65 in this study. This apparent deviation suggests that, despite increased overall oxygen supply, the conditions required for complete CO burnout were not met. In lab-scale reactors, increasing ER increases the total gas flow rate and can shorten the effective residence time of CO within the hot oxidation zone. Under these circumstances, a portion of CO generated in upstream reactions may exit the oxidation zone before being fully oxidized to $CO_2$, resulting in sustained or increasing CO concentrations even at higher ER. As a result, higher ER settings may be required for lab-scale gasifiers, which inherently have limited thermal retention.

The relatively low concentrations of $H_2$ and $CH_4$ further support the notion of temperature-limited reaction progress. $H_2$ is primarily formed through temperature-dependent reactions such as the water-gas shift ($CO + H_2O \leftrightarrow CO_2 + H_2$), steam reforming (e.g., $CH_4 + H_2O \leftrightarrow CO + 3H_2$), and tar reforming ($C_mH_n + mH_2O \rightarrow mCO + (m+n/2)H_2$). These reactions are significantly enhanced above 1000 °C [21,24–26]. Additionally, the use of charcoal, which contains less hydrogen than raw biomass, may have further limited $H_2$ production. A recent study reported that $H_2$ yield from charcoal prepared at 300 °C was lower than that from raw biomass at a gasification temperature of 600 °C; however, this difference largely disappeared at 1000 °C [9]. Therefore, the low gasification temperature remains the most plausible reason for the suppressed $H_2$ and $CH_4$ yields observed in this study.

The insufficient temperature rise may also be attributed to incomplete oxygen utilization. Although increasing ER typically correlates with higher temperatures, the relatively high $O_2$ concentrations observed in the syngas (5.9–8.3 vol%)

indicate that not all supplied oxygen was consumed. By contrast, a previous study at ER = 0.2 and gasification temperatures up to 1000 °C reported residual $O_2$ levels as low as 0.3–1.3 vol% [9]. One likely explanation is the formation of preferential flow channels between the auger and the reactor wall, which could have allowed air to bypass the feedstock, reducing air–fuel contact. The observed decline in $O_2$ concentration with increasing ER suggests that higher airflow enhanced turbulence and improved oxygen–feedstock mixing, although not sufficiently. The estimation of temperature rise due to further oxygen consumption will be discussed in a later section.

### 3.3. Product and elemental distribution

The distributions of product phases (syngas, char, and liquid) and the corresponding elemental carbon and hydrogen among these phases at different ERs are shown in Fig 4. While the product distribution shows the overall transformation behavior, the carbon and hydrogen distributions provide insight into the fate of individual elements during gasification.

Increasing the ER resulted in a higher proportion of gaseous products, accompanied by a reduction in char fraction (Fig 4a). At ER = 0.47, the syngas yield accounted for approximately 44%, while char and liquid constituted the remaining 56%. At ER = 0.60, the syngas fraction increased to over 60%, indicating enhanced conversion of solid and volatile components into gaseous products with greater oxygen availability. A slight increase in liquid products suggests that water vapor formation was enhanced as ER increased, likely due to higher oxidation activity and elevated temperatures.

The carbon distribution among the gas, liquid, and char phases is shown in Fig 4b. A similar trend was observed, with the proportion of carbon recovered in the gas phase increasing from approximately 46% at ER = 0.47 to 74% at ER = 0.60. This shift implies that a greater fraction of the carbon content in the biomass was converted into gaseous carbon species such as CO and $CO_2$ at higher ERs, reflecting improved CCE. Correspondingly, the carbon content in the char phase declined significantly, indicating more complete gasification of the solid matrix.

As shown in Fig 4c, hydrogen was predominantly distributed in the gaseous phase across all ERs. The gas-phase hydrogen proportion increased slightly with ER, from approximately 47% at ER = 0.47 to over 64% at ER = 0.60, while the hydrogen content in the liquid and solid phases decreased. This trend suggests that hydrogen-rich volatiles and water vapor were more effectively cracked and reformed into $H_2$ at higher ERs, facilitated by higher temperatures and the presence of oxidizing agents.

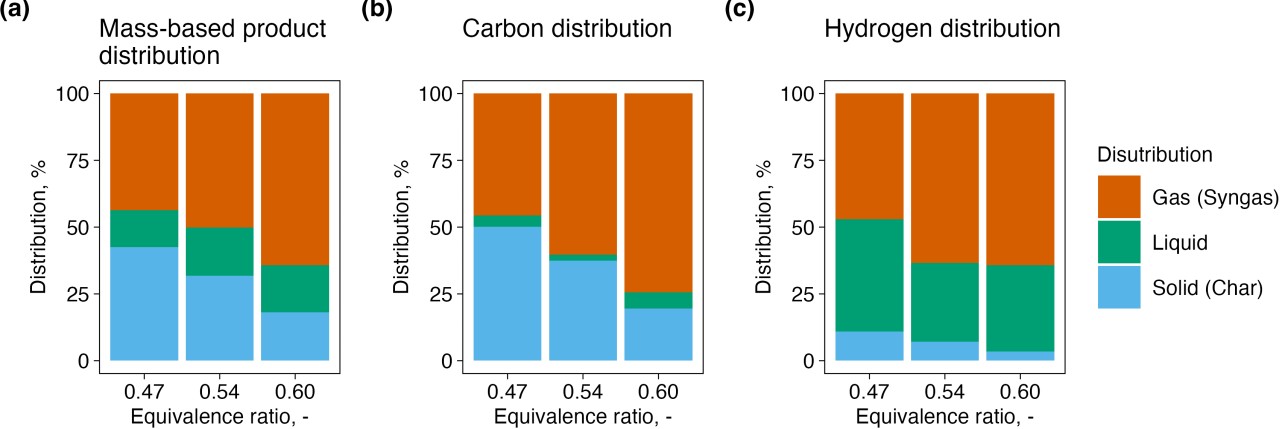

**Fig 4. Product and elemental (C, H) distributions among gas (syngas), liquid, and solid (char) fractions at different equivalence ratios.** (a) Product distribution on a mass basis relative to the total dry inputs (dry biomass + dry air) (solid moisture assumed to be zero; tar included in the liquid fraction). (b) Carbon distribution as a fraction of input carbon. (c) Hydrogen distribution as a fraction of input hydrogen.

These results illustrate that higher ER promotes the transformation of both mass and elements (C and H) into gaseous products, consistent with the trends observed in syngas composition, LHV, and CGE. The distributions also highlight the reduction of residual char and liquid byproducts, which is beneficial for achieving cleaner gasification and higher system efficiency. Thus, ER control plays a crucial role not only in energy conversion but also in influencing the fate of elements within the reactor.

### 3.4. LHV of product gas, CCE and CGE

As shown in Fig 5, the LHV, CCE, and CGE all exhibited an increasing trend with rising ER. This suggests that higher ER values enhance both the energy content and carbon utilization efficiency of the product gas. Specifically, the LHV increased from 1.90 MJ/Nm$^3$ at ER = 0.47 to 2.60 MJ/Nm$^3$ at ER = 0.60, primarily due to higher concentrations of CO and $H_2$, which are key energy carriers. While this shows a clear increasing trend with ER, it is still lower than the 4–6 MJ/Nm$^3$ range reported for other small-scale gasifiers [5–8]. This suggests that although higher ER contributes to LHV enhancement, the effects of incomplete oxygen utilization and heat loss in the compact system remains significant.

Similarly, the CGE increased from 26.6% to 48.4% over the same ER range, reflecting more effective energy recovery under elevated ER conditions. In parallel, the CCE, representing the fraction of carbon in the feedstock converted into gaseous products, rose from 45.0% to 74.4%. Although the target CGE of 80% was not achieved in this study, previous research has reported CGE and CCE values of approximately 80% and 90%, respectively, under conditions of ER ≈ 0.2 and a gasification temperature of 1000 °C [9]. These findings suggest that achieving the target efficiency is feasible if higher gasification temperatures can be realized. Therefore, future research should focus on improving reactor design to increase the internal reaction temperature, potentially enabling the system to reach CGE levels of 80% or more.

### 3.5. Thermal analysis

The gasification temperatures achieved in this study were significantly lower than those reported in prior literature, particularly in the reduction zone. While typical downdraft gasifiers operating at small to medium scales (50–100 kW$_{th}$) have demonstrated reduction zone temperatures exceeding 800–900 °C [16,18], the maximum temperature recorded in the present system was approximately 620 °C at T2 (ER = 0.60; i.e., the highest among the steady-state conditions tested). Two principal factors are considered to have contributed to this temperature suppression: incomplete oxygen consumption and insufficient thermal insulation.

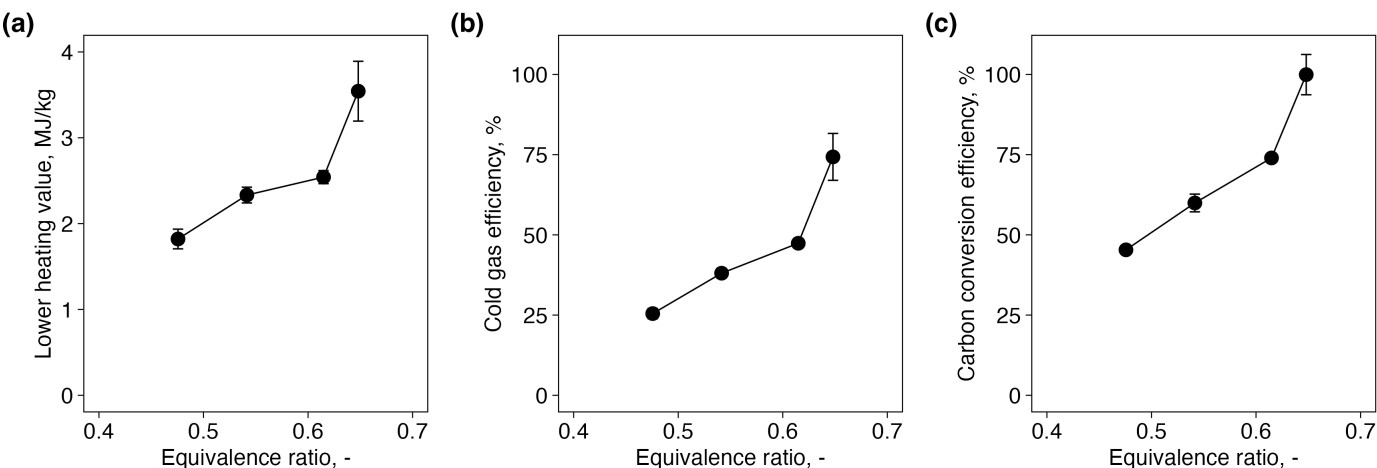

**Fig 5. Effects of equivalence ratio on (a) lower heating value, (b) cold gas efficiency, and (c) carbon conversion efficiency.**

First, we observed the relatively high residual oxygen concentrations in the product gas (5.9–8.3 vol%), as described in Section 3.2. This resulted in reduced combustion intensity and, consequently, limited heat release. An upper-bound thermodynamic estimation indicated that if oxygen were consumed to a level below 1 vol%, the theoretical maximum temperature could increase by 573 °C, potentially exceeding 1000 °C in oxidation zone. Such a temperature rise would significantly enhance syngas quality and conversion efficiencies, given the strong temperature dependence of key reactions such as the Boudouard reaction and steam reforming.

Second, the thermal insulation capacity of the reactor was likely insufficient to minimize heat loss under the experimental conditions. The outer wall was wrapped in a 15 mm-thick alumina blanket, which, despite its moderate thermal resistance, may not have been adequate under high temperature gradients. Consequently, a simplified energy balance analysis, based on the difference between the total reaction enthalpy (CO and $CO_2$ formation) and the sensible heat absorbed by the product gas, was conducted. This analysis revealed that under ER = 0.60 conditions, approximately 30.5% of the heat generated by oxidation reactions was lost to the surroundings. This loss rate is significantly higher than the ~3% and ~10% reported for similar systems [27,28]. These findings support the hypothesis that substantial heat loss associated with the small reactor scale limited the achievable temperature rise.

To mitigate such losses, improvements in reactor insulation—such as increasing the blanket thickness or using advanced multi-layer insulating materials—should be explored. Alternatively, scaling up the reactor diameter may reduce the surface-to-volume ratio, thereby decreasing the rate of conductive and radiative heat loss per unit mass of fuel. For instance, increasing the reactor diameter from 46 mm to 60 mm while maintaining similar operating parameters could theoretically reduce the heat loss flux and increase internal temperature by around 90 °C, indicating that gasification temperature can approach 1000 °C.

These results highlight that further improvements in reactor design—both in terms of internal flow dynamics and thermal control—are critical for achieving high-temperature gasification in ultra-small-scale systems. By ensuring more complete oxygen utilization and reducing heat losses, it may be possible to realize the thermodynamic conditions necessary for producing syngas of sufficiently high quality to drive downstream energy conversion systems such as SOFCs.

## 4. Conclusions

This study assessed the feasibility of achieving self-sustained operation in a laboratory-scale downdraft biomass gasifier without the use of external heat sources, aiming to support the development of household-scale CHP systems. The experimental results demonstrated that stable operation was attainable when the ER was controlled within the range of 0.47 to 0.60. Under these conditions, the reactor maintained a stratified temperature profile, enabling consistent gasification performance.

Although the product gas exhibited increasing concentrations of CO and $H_2$ with rising ER, the overall gas quality, LHV, CCE, and CGE remained below those reported in studies involving larger-scale systems. These discrepancies were primarily attributed to lower internal temperatures, arising from intensified heat losses and incomplete oxygen utilization likely caused by preferential flow channels within the reactor. Nonetheless, the observed enhancements in gas yield and elemental conversion with increasing ER indicate that operational optimization can partially compensate for such limitations.

Overall, the findings confirm the technical feasibility of self-sustained biomass gasification at a laboratory scale. To further improve efficiency and gas quality, future research should focus on advancing reactor insulation, optimizing air–fuel mixing, and redesigning flow pathways to enhance thermal retention. In addition, staged air injection, syngas recirculation, and catalytic enhancements could be explored to intensify high-temperature reaction zones and promote secondary reactions (e.g., tar cracking/reforming), thereby improving syngas composition and overall gasification efficiency. These efforts will be critical for scaling down gasification technology toward practical applications in ultra-small-scale, decentralized, carbon-neutral energy systems.

## Supporting information

**S1 Table. Molar fraction and molar heat capacities of the syngas components at ER = 0.60, used for calculating the mixture heat capacity.**
(DOCX)

## Acknowledgments

The authors would like to thank the Instrumental Analysis Division of Hokkaido University for conducting elemental analysis using a CE-440 elemental analyzer.

## Author contributions

**Conceptualization:** Takanori Itoh, Kazunori Iwabuchi.

**Formal analysis:** Takanori Itoh, Shota Hanabusa.

**Funding acquisition:** Kazunori Iwabuchi.

**Investigation:** Shota Hanabusa.

**Supervision:** Kazunori Iwabuchi.

**Writing – original draft:** Takanori Itoh.

**Writing – review & editing:** Kazunori Iwabuchi.

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
