## [Decision Letter · Decision Letter 0]

11 Jan 2026

Dear Dr. Iwabuchi,

Thank you for submitting your manuscript to PLOS ONE. After careful consideration, we feel that it has merit but does not fully meet PLOS ONE’s publication criteria as it currently stands. Therefore, we invite you to submit a revised version of the manuscript that addresses the points raised during the review process.

We look forward to receiving your revised manuscript.

Kind regards,

Debajyoti Kundu, Ph.D

Academic Editor

PLOS One

**Journal Requirements:**

“This study was supported by funding from Tanigurogumi Corporation, Japan, and the JSPS KAKENHI (grant number JP23K23727).”

We note that you received funding from a commercial source: Tanigurogumi Corporation

3. Please expand the acronym “JSPS KAKENHI” (as indicated in your financial disclosure) so that it states the name of your funders in full.

[“This study was supported by funding from Tanigurogumi Corporation, Japan, and the JSPS KAKENHI (grant number JP23K23727).”

5. Please upload a new copy of Figures 1, 2, 3, 4, and 5 as the detail is not clear. Please follow the link for more information:  https://journals.plos.org/plosone/s/figures

6. Please include captions for your Supporting Information files at the end of your manuscript, and update any in-text citations to match accordingly. Please see our Supporting Information guidelines for more information: http://journals.plos.org/plosone/s/supporting-information .

**Additional Editor Comments:**

The reviewers have recommended **minor revisions**  to improve clarity and presentation of the manuscript. Please address all comments carefully and submit a revised version along with a point-by-point response explaining how each suggestion has been addressed.

Reviewers' comments:

Reviewer's Responses to Questions

**Comments to the Author**

1. Is the manuscript technically sound, and do the data support the conclusions?

Reviewer #1: Yes

Reviewer #2: Yes

2. Has the statistical analysis been performed appropriately and rigorously?

Reviewer #1: Yes

Reviewer #2: Yes

3. Have the authors made all data underlying the findings in their manuscript fully available?

Reviewer #1: Yes

Reviewer #2: Yes

4. Is the manuscript presented in an intelligible fashion and written in standard English?

Reviewer #1: Yes

Reviewer #2: Yes

Reviewer #1: The work presents an interesting and technically sound investigation into the feasibility of self-sustained biomass gasification at the laboratory scale, specifically within a very compact 46 mm downdraft reactor. It directly addresses the challenge of downscaling biomass systems for household-level combined heat and power (CHP) applications, and I strongly believe that it contributes new insights into how oxygen utilization and thermal management control the performance of such small reactors.

The experiments are generally well designed, with clear reporting of fuel properties, system design, and operational parameters. The data support the main conclusions: self-sustained operation was achieved in the ER range of 0.47- 0.60, gas composition improved with increasing ER, and yet performance (in terms of CO and H₂ concentrations, LHV, CCE, and CGE) lagged behind values reported in larger-scale studies. The explanation given, that heat losses and incomplete oxygen consumption suppressed temperatures in the reduction zone, is consistent with both the presented results and prior literature.

That said, there are several points where the manuscript could be strengthened: For instance,

1. Figures are clear and informative, and the narrative is well structured. However, I recommend that in Figure 3 and Figure 5 (a, b and c), the error bars be shown only in the y-direction, with the x-component removed, to provide a clearer presentation.

2. On the Gas composition discussion, the observed CO and H₂ concentrations are notably below those typically reported. The manuscript attributes this mainly to temperature suppression and oxygen bypass. This explanation is sound, but a deeper discussion of possible effects such as short residence times and channeling in the bed would provide a fuller picture of the gasifier limitations.

3. From the Quantification of the heat loss, the estimate that ~30.5% of the generated heat was lost is critical to the conclusions. At present, much of the supporting analysis is relegated to the Supplementary Information. A clearer presentation of the calculation method and assumptions in the main text would enhance transparency.

4. The discussion already points toward improvements in insulation and mixing. It would strengthen the conclusion to briefly acknowledge other potential approaches that could mitigate temperature suppression, such as staged air injection, syngas recirculation, or catalytic enhancements. Even a short mention would situate the work more firmly within the broader trajectory of small-scale gasifier development. This could be addressed as a perspective or strategy that future designs could look out for.

5. The manuscript is generally well written and easy to follow, but a few sentences could be polished for grammar and clarity (for example, “self-sustaining operation … is capable” could be revised to “is achievable”).

In summary, this is a valuable contribution that demonstrates the feasibility of lab-scale autothermal gasification and highlights the central role of oxygen utilization and thermal retention at small scales. With minor revisions to strengthen data presentation, expand on the discussion of gas composition and heat losses, and polish the language, this work will be suitable for publication.

Reviewer #2: This manuscript presents a well-structured experimental study on feasibility of achieving autothermal gasification in an lab scale downdraft reactor without external heating stable operation is possible despite the severe heat losses typical of small-scale reactors. The study is novel and sound and well written. Minor Revisions can be done.

Line 236: Authors present a interesting finding that diverges from established literature, Higher ER also produces higher CO thus translating to higher CCV and LHVs. Authors should also discuss in the context of the higher ER, oxidation zone is too small or the residence time is too short for the CO to be fully oxidized, despite the high oxygen availability?

Line 272: Revise figure 4 caption for more clarity.

Line 339: Authors claim the maximum T2 obtained is 620 °C but in Fig 2d T2 spikes much further at 30 min. needs further discussion. Consider revising the paragraph

**Do you want your identity to be public for this peer review?** For information about this choice, including consent withdrawal, please see our Privacy Policy

Reviewer #1: No

Reviewer #2: No

---

## [Author Response · Author response to Decision Letter 1]

27 Jan 2026

Resonse to Reviewrs Comments

We thank the reviewers for their thoughtful and supportive comments. We have revised our manuscript in response to their suggestions and hope that this improved manuscript is acceptable for publication in PLOS ONE.

Response to Reviewer #1

The work presents an interesting and technically sound investigation into the feasibility of self-sustained biomass gasification at the laboratory scale, specifically within a very compact 46 mm downdraft reactor. It directly addresses the challenge of downscaling biomass systems for household-level combined heat and power (CHP) applications, and I strongly believe that it contributes new insights into how oxygen utilization and thermal management control the performance of such small reactors.

The experiments are generally well designed, with clear reporting of fuel properties, system design, and operational parameters. The data support the main conclusions: self-sustained operation was achieved in the ER range of 0.47-0.60, gas composition improved with increasing ER, and yet performance (in terms of CO and H2 concentrations, LHV, CCE, and CGE) lagged behind values reported in larger-scale studies. The explanation given, that heat losses and incomplete oxygen consumption suppressed temperatures in the reduction zone, is consistent with both the presented results and prior literature.

That said, there are several points where the manuscript could be strengthened: For instance,

Comment 1:

Figures are clear and informative, and the narrative is well structured. However, I recommend that in Figure 3 and Figure 5 (a, b and c), the error bars be shown only in the y-direction, with the x-component removed, to provide a clearer presentation.

Response: As recommended, we revised Figure 3 and Figure 5 (a–c) so that the error bars are displayed only in the y-direction, and the x-direction error bars have been removed to improve clarity and readability.

Comment 2:

On the Gas composition discussion, the observed CO and H2 concentrations are notably below those typically reported. The manuscript attributes this mainly to temperature suppression and oxygen bypass. This explanation is sound, but a deeper discussion of possible effects such as short residence times and channeling in the bed would provide a fuller picture of the gasifier limitations.

Response: Thank you for this helpful suggestion. We agree that, in addition to temperature suppression and oxygen bypass, hydrodynamic limitations such as short effective residence time and channeling can further constrain syngas formation in lab-scale fixed-bed gasifiers. Accordingly, we expanded the discussion in Section 3.2 (Gas composition) to explicitly address (i) reduced effective residence time of volatiles/product gas in the high-temperature reaction zone due to limited reactor volume and preferential flow (channeling) and (ii) how these limitations can suppress H2 formation via limited secondary reforming/tar cracking and limit additional CO generation via insufficient gas–solid reaction progress (e.g., char–CO2/steam reactions). These additions provide a fuller explanation for why the CO and H2 concentrations obtained in this study are lower than those typically reported.

Chage in manuscript: Lines 294–303: In addition to temperature limitation, lab-scale fixed-bed systems can exhibit reduced effective residence time of volatiles and product gas within the high-temperature reaction zone due to limited reactor volume and preferential flow (channeling), which constrains secondary gas-phase reactions (e.g., tar cracking/reforming) and gas–solid reactions [22, 23]. In particular, limited secondary reforming/tar cracking can suppress H2 formation, while insufficient gas–solid reaction progress (e.g., char–CO2/steam reactions) can limit additional CO generation. Such hydrodynamic limitations could also contribute to the comparatively low CO and H2 concentrations observed here, even when the maximum bed temperature approaches ~1000 °C.

Comment 3:

From the Quantification of the heat loss, the estimate that ~30.5% of the generated heat was lost is critical to the conclusions. At present, much of the supporting analysis is relegated to the Supplementary Information. A clearer presentation of the calculation method and assumptions in the main text would enhance transparency.

Response: We appreciate the reviewer’s comment. We have transferred the heat-loss calculation procedure and key assumptions from the Supplementary Information to the Methods section (Section 2.5, Thermal analysis and energy-balance calculations), including the definition of the heat-loss fraction based on the overall energy balance. The Supporting Information now contains only supporting property table (S1 Table).

Comment 4:

The discussion already points toward improvements in insulation and mixing. It would strengthen the conclusion to briefly acknowledge other potential approaches that could mitigate temperature suppression, such as staged air injection, syngas recirculation, or catalytic enhancements. Even a short mention would situate the work more firmly within the broader trajectory of small-scale gasifier development. This could be addressed as a perspective or strategy that future designs could look out for.

Response: We appreciate the reviewer’s suggestion. In the Conclusions section, we have added a brief perspective acknowledging additional strategies to further improve gas quality and gasification efficiency in ultra-small-scale systems.

Chage in manuscript: Lines 471–474: In addition, staged air injection, syngas recirculation, and catalytic enhancements could be explored to intensify high-temperature reaction zones and promote secondary reactions (e.g., tar cracking/reforming), thereby improving syngas composition and overall gasification efficiency.

Comment 5:

The manuscript is generally well written and easy to follow, but a few sentences could be polished for grammar and clarity (for example, “self-sustaining operation … is capable” could be revised to “is achievable”).

Response: We thank the reviewer for this comment. We have carefully proofread the manuscript and revised several sentences for improved grammar and clarity, including changing “self-sustaining operation … is capable” to “self-sustaining operation … is achievable.”

In summary, this is a valuable contribution that demonstrates the feasibility of lab-scale autothermal gasification and highlights the central role of oxygen utilization and thermal retention at small scales. With minor revisions to strengthen data presentation, expand on the discussion of gas composition and heat losses, and polish the language, this work will be suitable for publication.

Response to Reviewer #2

This manuscript presents a well-structured experimental study on feasibility of achieving autothermal gasification in an lab scale downdraft reactor without external heating stable operation is possible despite the severe heat losses typical of small-scale reactors. The study is novel and sound and well written. Minor Revisions can be done.

Comment 6:

Line 236: Authors present a interesting finding that diverges from established literature, Higher ER also produces higher CO thus translating to higher CCV and LHVs. Authors should also discuss in the context of the higher ER, oxidation zone is too small or the residence time is too short for the CO to be fully oxidized, despite the high oxygen availability?

Response: Thank you for pointing this out. We acknowledge that the increasing CO trend with ER contrasts with the commonly reported decrease in CO at higher ER due to enhanced oxidation. To address this, we expanded the discussion in Section 3.2 (Gas composition) to explain that, in lab-scale reactors, increasing ER increases the total gas flow rate and may shorten the effective residence time of CO within the hot oxidation zone. Under such conditions, CO generated upstream can leave the oxidation zone before complete conversion to CO2, leading to sustained or even increased CO concentrations despite increased overall oxygen supply. This additional discussion clarifies a plausible mechanism behind the observed deviation from established trends.

Chage in manuscript: Lines 307–310: This apparent deviation suggests that, despite increased overall oxygen supply, the conditions required for complete CO burnout were not met. In lab-scale reactors, increasing ER increases the total gas flow rate and can shorten the effective residence time of CO within the hot oxidation zone. Under these circumstances, a portion of CO generated in upstream reactions may exit the oxidation zone before being fully oxidized to CO2, resulting in sustained or increasing CO concentrations even at higher ER.

Comment 7:

Line 272: Revise figure 4 caption for more clarity.

Response: We have revised the caption of Fig. 4 to improve clarity, as suggested.

Chage in manuscript: Lines 345–349: Fig. 4. Product and elemental (C, H) distributions among gas (syngas), liquid, and solid (char) fractions at different equivalence ratios. (a) Product distribution on a mass basis relative to the total dry inputs (dry biomass + dry air) (solid moisture assumed to be zero; tar included in the liquid fraction). (b) Carbon distribution as a fraction of input carbon. (c) Hydrogen distribution as a fraction of input hydrogen.

Comment 8:

Line 339: Authors claim the maximum T2 obtained is 620 °C but in Fig 2d T2 spikes much further at 30 min. needs further discussion. Consider revising the paragraph

Response: Thank you for this comment. As the reviewer noted, T2 in Fig. 2d (ER = 0.65) showed a transient spike around 30 min, which exceeds 620 °C. However, under ER = 0.65, combustion became dominant and steady-state operation could not be achieved because the feedstock bed height could not be maintained, as described in the first paragraph of Section 3.1. We therefore judged this condition to have poor reproducibility and excluded it from the thermal analysis.

In the revised manuscript, References 15, 22, and 23 have been newly added.

---

## [Editor Report · Decision Letter 1]

8 Feb 2026

Realizing self-sustained biomass gasification in a lab-scale downdraft reactor for compact CHP applications

PONE-D-25-41582R1

Dear Dr. Iwabuchi,

We’re pleased to inform you that your manuscript has been judged scientifically suitable for publication and will be formally accepted for publication once it meets all outstanding technical requirements.

Kind regards,

Debajyoti Kundu, Ph.D

Academic Editor

PLOS One

Additional Editor Comments (optional):

accept
---

## [Editor Report · Acceptance letter]

PONE-D-25-41582R1

PLOS One

Dear Dr. Iwabuchi,

I'm pleased to inform you that your manuscript has been deemed suitable for publication in PLOS One. Congratulations! Your manuscript is now being handed over to our production team.

Kind regards,

on behalf of

Dr. Debajyoti Kundu

Academic Editor

PLOS One